# Feasibility Study for a Python-Based Embedded Real-Time Control System

Se Yeon Cho [1] , Raimarius Delgado [2] and Byoung Wook Choi [1,*]

1 Department of Electrical and Information Engineering, Seoul National University of Science and Technology, Seoul 01811, Republic of Korea; seyeon@seoultech.ac.kr
2 Center for Intelligent & Interactive Robotics, Korea Institute of Science and Technology, Seoul 02792, Republic of Korea; raim.delgado@kist.re.kr
* Correspondence: bwchoi@seoultech.ac.kr; Tel.: +82-02-970-6412

**Abstract:** Because of its simplicity and the support of numerous useful libraries, Python has become one of the most popular programming languages for application development, even in embedded systems. However, in existing control systems where specific tasks must meet specific temporal deadlines and support schedulability with proper priority assignments, the Python interpreter may not satisfy real-time requirements, owing to features such as the global interpreter lock and garbage collector. This paper addresses these constraints with an approach that executes periodic real-time tasks under the fixed-priority preemptible scheduler of RT-Preempt. First, we implemented a Python real-time module that allows users to create and execute periodic tasks with fixed priorities based on Python. Then, we conducted experiments on an open embedded system, in this case, a Raspberry Pi 4. We evaluated the real-time performance, focusing on test metrics for control systems, such as task periodicity, responsiveness, and interrupt response. The results were then compared to those of conventional real-time tasks developed using the C language to validate the feasibility of the proposed method. Finally, we performed experimental validation by tracking the position of EtherCAT servo motors to demonstrate the feasibility of a Python-based real-time control system in a practical application.

**Keywords:** Python; embedded systems; real-time; RT-Preempt; EtherCAT





## 1. Introduction

Python is an object-oriented, interpreted language that has gained popularity from its easy access to many development packages, such as Numpy, OpenCV, TensorFlow, and PyQT. Python also includes well-known modules for machine learning and scientific computing [1,2]. In addition, with a very active community, Python has become the most preferred programming language for project development over the last few years, ranking first in the annual interactive rankings of IEEE Spectrum [3].

However, Python has disadvantages regarding speed and capacity, making it challenging to apply in embedded environments with limited computing power and resources. Some researchers have proposed developing Python-based applications in embedded fields, such as the IoT [4]. Still, these efforts need to address the real-time requirements of embedded systems. Herein, real time refers to the ability to execute real-time tasks bounded by hard temporal deadlines to ensure deterministic and predictable responses of the entire system [5]. Real-time constraints should be considered to avoid any task failure, which can result in system faults or, worse, physical damage and accidents [6,7].

Real-time operating systems (RTOSs) are mainly employed to develop real-time applications. RTOSs mainly offer low-level APIs available in C/C++ languages, owing to their predictable runtime performance essential for reliable real-time execution. On the other hand, an interpreted language, such as Python, is assumed unsuitable to meet these

demands because of its global interrupt lock and garbage collection processes [8]. There have been attempts to improve the runtime speed of Python. For example, NumPy, short for Numerical Python, integrates C/C++ and Fortran to execute vector and matrix operations without calling back into Python. In addition, Numba compiles Python code and runs a faster version inside the typical interpreter runtime. Still, more is needed to meet the temporal deadlines required in real-time systems.

Ionescu et al. [9] reported that MicroPython could be an alternative to C/C++ to reduce the complexity of developing applications for embedded systems, employing MicroPython in real-time systems. However, it still requires further resource management and garbage collection optimizations. For example, Bucher et al. [10] developed the pysimCoder package to design control systems in Python and generate the C/C++ code to be compiled for real-time application. The common drawback of these studies is their limited—or complete absence of—support for applications with multiple tasks (threads).

This paper describes a Python method for executing real-time periodical tasks under the real-time Linux extension RT-Preempt using a POSIX Linker and the Ctypes module [11,12]. The method is tested on a Raspberry Pi 4 device and evaluated through various real-time performance analyses, including periodicity and interrupt latency [13]. The results are compared to those of the same program developed in C/C++. The paper also demonstrates the feasibility of a Python-based real-time embedded control system by performing servo motor control using the industrial Fieldbus EtherCAT [14–17]. This work aims to provide a stepping-stone for easier integration of recent trends in machine learning and numerical analysis on real-time systems for use in various fields, such as data analytics, robotics, and industrial control. The main contributions of this study are as follows:

- The development of a Python method for executing real-time periodical tasks under the real-time Linux extension RT-Preempt, using a POSIX Linker and the Ctypes module;
- An evaluation of the method on a Raspberry Pi 4 device through various real-time performance analyses, including periodicity and interrupt latency;
- A comparison of the results to those of the same program developed in C/C++ to validate the potential of the Python-based real-time system;
- A demonstration of the feasibility of a Python-based real-time embedded control system by performing servo motor control using the industrial Fieldbus EtherCAT;
- Providing a stepping-stone for easier integration of recent trends in machine learning and numerical analysis on real-time systems for use in various fields, such as data analytics, robotics, and industrial control.

The rest of this document consists of the following: Section 2 presents and implements the structure and implementation of Python-based real-time systems. Furthermore, experiments are described and conducted to understand the performance and feasibility of the proposed Python-based real-time system. The results of the experiment are then presented and discussed. Section 3 uses EtherCAT to construct and experiment with systems for real-time control of servo motors. The results also demonstrate the feasibility of Python-based real-time systems presented in the study through analysis via probabilistic statistical methods. Finally, Section 4 summarizes our findings and discusses future research.

## 2. Python-Based Real-Time System

This chapter describes the development environment and system configuration used to implement the Python-based EtherCAT master and the architecture of the Python-based EtherCAT master.

### 2.1. Python-Based Real-Time System Architecture

In this paper, Python-based embedded real-time systems are implemented using RT-Preempt patched Linux kernel 5.10.110-rt63, capable of low-latency scheduling, and the Debian 11 Bullseye file system. The real-time system architecture is described in Figure 1. It consists of the hardware (H/W), operating system (OS), middleware, and task layers.

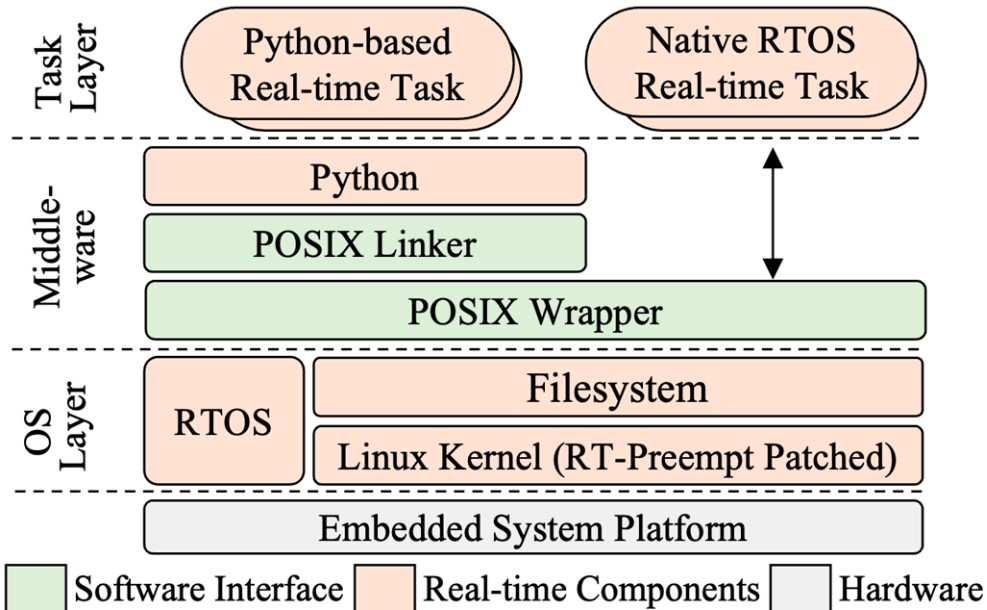

**Figure 1.** Python-based real-time system architecture.

The H/W layer denotes the hardware platform on which the system is mounted. The OS layer consists of the kernel and other development environment primitives, such as the filesystem, RTOS, and APIs. The middleware layer is directly exposed to the task layer, consisting of POSIX Wrapper, POSIX Linker, and Python. Its primary responsibility is to wrap implementation into general user-space APIs so users can easily develop real-time python applications without knowledge of the system internals.

The task layer in user space consists of tasks operating under real-time constraints and can be implemented on a Python or native.

### 2.2. Interface Real-Time Library to Python

This paper uses POSIX Linker functions to extend Python's real-time features. POSIX Wrapper is a derivative of an ongoing project to develop an all-in-one solution for real-time Linux environments with ideas such as RT-AIDE [18]. It is a C/C++-based real-time dynamic link library inspired by popular RTOSs, such as FreeRTOS and Xenomai, implemented using POSIX-API, which uses POSIX threads to create real-time tasks easily, manage real-time clocks and timers, and manage other interaction mechanisms, including mutex, message queues, and semaphores. It is also a quality-controlled one using Google Test (GTest) and Jenkins through unit testing, code coverage, and static analysis monitoring.

POSIX Linker is the interface between the POSIX Wrapper and Python based on the Ctypes package. Using Ctypes, the POSIX Wrapper objects and functions can be accessed natively in Python. Table 1 shows the functions of POSIX Wrapper and its respective POSIX Linker in Python.

**Table 1.** Functions to Create Real-time Tasks Using POSIX Wrapper and their Respective POSIX Linker.

| Functionality | POSIX Wrapper | POSIX Linker |
|---|---|---|
| Task Handler | POSIX_TASK | PY_POSIX_TASK |
| Task Creation | create_rt_task | py_create_rt_task |
| Start Execution | start_task | py_start_task |
| Set Task Timer | set_task_period | py_set_task_period |
| Wait Period | wait_next_period | py_wait_next_period |

Task Handler consists of information for controlling and managing tasks, such as process ID, name, stack size, period, and priority. Task creation generates a task handler and stores the period, priority, stack size, and name information that are user-specified arguments. In Start Execution, storing the user-typed function in a task handler, configuring the task as the previously stored task information, and executing the user function are conducted. Set Task Timer can store or change the task handler period information. Wait Period is a time wait function that has an important function in creating periodicity, one of the real-time features. This should be called within the function of a periodic task to wait for the next scheduled entry point after the current iteration has expired. Read Time is a function that reads the current system time, and the unit is nanosecond.

### 2.3. Performance in a Multi-Tasking Environment

This chapter compares the results of performing the same operation implemented on C/C++ and Python to identify and validate the potential of the proposed Python-based real-time system.

The platform and system configuration used for the experiment is as introduced in Figure 1. In addition, we focused on two performance metrics determining stability and reliability in real-time control applications.

First, we evaluated the periodicity and responsiveness to determine if we could perform the task while meeting the specified deadline.

Second, we conducted an experiment to measure the interrupt response time. This demonstrated the performance capabilities of embedded devices when interacting with digital input/output device drivers. Finally, experimental data measurements were obtained via software timing probes and through hardware to analyze actual system behavior using an oscilloscope.

In addition, to build confidence that all of Python's real-time extensions are suitable for real-time applications, we measured the scheduling latency of each extension under the influence of the interference load. In this respect, for example, we considered the two best loads: CPU and memory. The interference load under consideration corresponds to the CPU and memory load utilization of the stress-ng tool [19].

The stress-ng was configured to run multiple floating-point arithmetic operations in a very tight infinite loop to simulate loads fully utilizing the CPU. A virtual memory stress test is also performed. With the memory requirement for storing observed values, memory stress can only occupy 70% of the system memory during the experiment.

#### 2.3.1. Periodicity and Responsiveness

The periodicity of a real-time system means that each task is scheduled to operate correctly at the predicted time, and responsiveness means that all tasks must be able to run within a certain period. This can verify the task analysis real-time systems by measuring both the period and worst response times [20].

The first experiment verifies the periodicity and responsiveness of each task among several tasks and confirms that priority-based preemption occurs to understand the potential of Python-based real-time systems. The task configuration for the experiment is shown in Table 2 and uses three tasks with different periods, deadlines, execution times, and priorities.

**Table 2.** Experiment Task Configuration.

| Task | Period (ms) | Deadline (ms) | Execution (ms) | Priority |
|------|-------------|---------------|----------------|----------|
| $\tau_1$ | 10 | 10 | 3 | 99 |
| $\tau_2$ | 20 | 20 | 5 | 80 |
| $\tau_3$ | 40 | 40 | 10 | 70 |

In RT-Preempt, 99 denotes the highest priority, and 1 is the lowest, and these three tasks operate on an isolated CPU core 1. Low priority $\tau_3$ is configured to run every 40 ms,

medium priority $\tau_2$ runs every 20 ms, and highest priority $\tau_1$ runs every 10 ms. As real-time tasks are scheduled with fixed-priority and pre-emptive scheduling, the worst-case response time (WCRT) can be calculated via rate monotonic analysis (RMA) [20].

$$R_i^{k+1} = C_i + B_i + \sum_{j \in hp(i)} \left\lceil \frac{R_i^k}{P_j} \right\rceil C_j \tag{1}$$

Herein, *C* denotes the execution time, and B denotes the blocking time, which is the time duration when a lower-priority task blocks higher-priority ones. The blocking time is zero for all tasks as long as RT-Preempt is correctly configured and no locking mechanisms are shared between the tasks. $hp(i)$ is the set of all tasks with higher priorities than the current task. The calculation for the WCRT requires the iteration of (1) until $R_i^{k+1} = R_i^k$ or $R_i^{k+1} \geq D_i$ is satisfied. Take note that for the first iteration, the response time is equal to the execution time. For the given real-time tasks, the WCRTs are calculated to be 3 ms, 8 ms, and 29 ms, respectively.

In the configured tasks, the function loop is conducted on each task to perform two performance metric measurements, such as actual period and response time. To simulate the computational load and the configured run time, we burn CPU resources for 1 ms and implement loop functions that are repeated until the required run time is completed. We call this function spin. In addition, each task is configured to toggle the GPIO output pin during every spin operation to measure the preemption and visualization through the oscilloscope.

The experiment uses wiringPi for the GPIO control of Raspberry Pi 4, and measurements are conducted for ten minutes to obtain sufficient sample measurement data. In addition, both C/C++ and Python languages, used to implement real-time tests in RT-Preempt, are used here to compare and analyze the periodicity and responsiveness during the Python real-time tests through statistical measurements.

Figures 2 and 3 show the results after measuring the periodicity by implementing real-time tests using Python and C/C++, respectively, and after measuring actual GPIO signals using an oscilloscope.

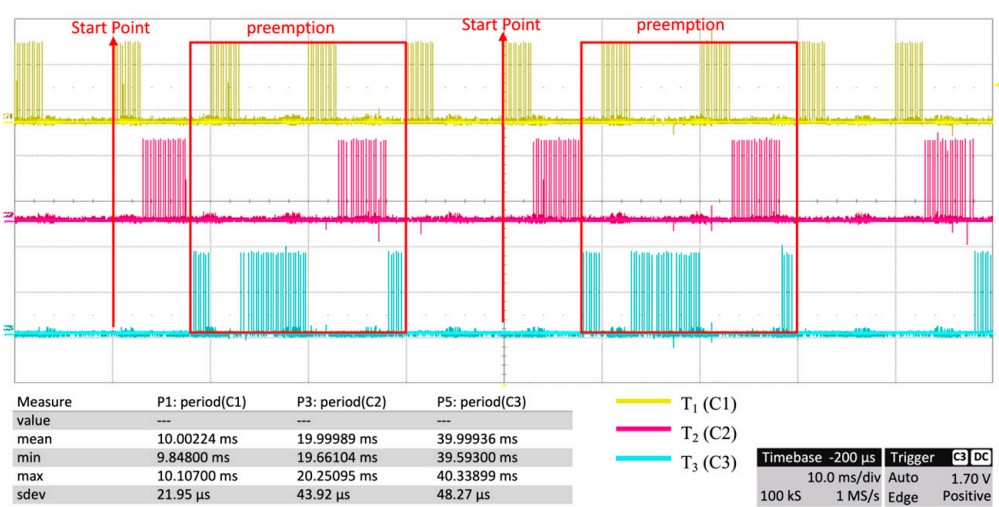

| Measure | P1: period(C1) | P3: period(C2) | P5: period(C3) |
| --- | --- | --- | --- |
| value | --- | --- | --- |
| mean | 10.00224 ms | 19.99989 ms | 39.99936 ms |
| min | 9.84800 ms | 19.66104 ms | 39.59300 ms |
| max | 10.10700 ms | 20.25095 ms | 40.33899 ms |
| sdev | 21.95 µs | 43.92 µs | 48.27 µs |

**Figure 2.** Real−time task periodicity using the oscilloscope in Python.

The figure of the waveforms uses a method of toggling GPIO when the task is running. It can be observed that for a single hyper-period, $\tau_1$ runs four times, and $\tau_2$ runs twice, while $\tau_3$ only has a single execution. A fixed-priority pre-emptive scheduler of RT-Preempt handles the execution of real-time tasks. Thus, it can be seen that $\tau_3$ is being pre-empted by $\tau_1$ and $\tau_2$ at specific points of its execution. The results show that the Python real-time test implementations also have priority preemption.

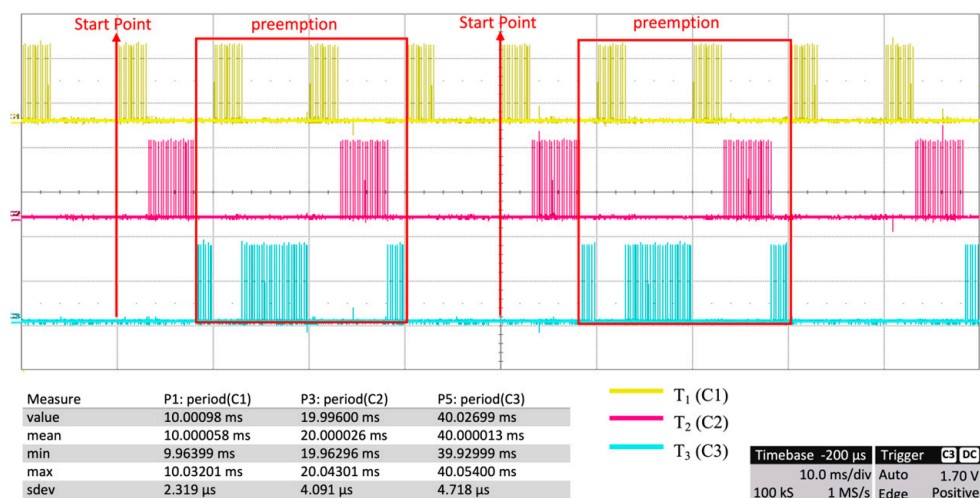

**Figure 3.** Real−time task periodicity using the oscilloscope in C/C++.

Table 3 shows the timing measurements of periodicity and response time in an idle environment without stress. In the period metric, both the C/C++ and Python implementations meet the mean duration of $\tau_1$, $\tau_2$, and $\tau_3$. Regarding the maximum values, it can be seen that the Python implementation has higher values compared to C/C++, with 10.088 ms, 20.222 ms, and 40.312 ms, respectively, for the real-time tasks. These slight differences are negligible relative to the total number of data samples and the minimal change in the standard deviation (Sdev).

**Table 3.** Period and Response time of Real-Time Task in idle environment.

| Metric | | C/C++ | | |
|---|---|---|---|---|
| | | $\tau_1$ | $\tau_2$ | $\tau_3$ |
| Period [ms] | Mean | 10.000 | 20.000 | 40.000 |
| | Min | 9.954 | 19.948 | 39.949 |
| | Max | 10.048 | 20.051 | 40.051 |
| | Sdev | 0.003 | 0.004 | 0.004 |
| Response [ms] | Mean | 3.008 | 8.017 | 29.021 |
| | Min | 3.006 | 8.014 | 29.019 |
| | Max | 3.056 | 8.067 | 29.068 |
| | Sdev | 0.002 | 0.003 | 0.003 |
| Metric | | Python | | |
| | | $\tau_1$ | $\tau_2$ | $\tau_3$ |
| Period [ms] | Mean | 10.000 | 20.000 | 40.000 |
| | Min | 9.918 | 19.783 | 39.693 |
| | Max | 10.088 | 20.222 | 40.312 |
| | Sdev | 0.023 | 0.038 | 0.013 |
| Response [ms] | Mean | 3.064 | 8.136 | 29.184 |
| | Min | 3.041 | 8.106 | 29.135 |
| | Max | 3.190 | 8.399 | 29.441 |
| | Sdev | 0.016 | 0.020 | 0.013 |

The response measurement results confirm that each task meets the requirement of the WCRT. Hence, it is possible to know whether the task operates periodically and accurately in a real-time system that is scheduled based on a fixed priority. It is confirmed that the WCRT expected by RMA is 3 ms at $\tau_1$, 8 ms at $\tau_2$, and 29 ms at $\tau_3$, and both C/C++ and Python satisfy the average value. However, at the maximum value, $\tau_2$ is 8.399 ms in Python, and $\tau_3$ is 29.441 ms, which is not satisfactory. However, if Sdev is checked, it can be seen

that $\tau_2$ is 0.020 ms and $\tau_3$ is 0.013 ms, which satisfies WCRT with a very low probability of occurrence.

Table 4 shows the results of the experiments in a stressed environment. To evaluate the behavior under interfering loads, the same experimental procedures are conducted with the following stress conditions: stress-ng utilizing 100% of the CPU and 70% of system memory. This method emulates an environment where multiple non–real-time threads attempt to occupy the CPU and memory resources when they are available. This test should have minimal effect on the performance of real-time tasks with high priorities. Otherwise, it can be concluded that real-time tasks violate real-time constraints, which may lead to missing stringent deadlines.

**Table 4.** Period and Response time of Real-Time Task in stress environment.

| Metric | | C/C++ | | |
|---|---|---|---|---|
| | | $\tau_1$ | $\tau_2$ | $\tau_3$ |
| | Mean | 10.000 | 19.999 | 40.000 |
| Period | Min | 9.935 | 19.929 | 39.935 |
| [ms] | Max | 10.068 | 20.068 | 40.058 |
| | Sdev | 0.009 | 0.014 | 0.010 |
| | Mean | 3.018 | 8.035 | 29.041 |
| Response | Min | 3.007 | 8.018 | 29.028 |
| [ms] | Max | 3.082 | 8.100 | 29.101 |
| | Sdev | 0.007 | 0.008 | 0.007 |
| Metric | | Python | | |
| | | $\tau_1$ | $\tau_2$ | $\tau_3$ |
| | Mean | 10.000 | 20.000 | 40.000 |
| Period | Min | 9.715 | 19.646 | 39.702 |
| [ms] | Max | 10.262 | 20.298 | 40.282 |
| | Sdev | 0.064 | 0.082 | 0.066 |
| | Mean | 3.216 | 8.474 | 29.636 |
| Response | Min | 3.060 | 8.219 | 29.287 |
| [ms] | Max | 3.459 | 8.751 | 29.871 |
| | Sdev | 0.066 | 0.067 | 0.069 |

In the results, it has been shown that the average period of all real-time tasks is equal to the idle environment with a minimal increase in the standard deviation. The response time also has shown a slight increase in comparison to the results in an idle environment. Even with their increased response times, all of the real-time tasks have met their respective deadlines. This means that Python-based real-time tasks with high priorities are scheduled as expected and are able to satisfy real-time constraints even in an environment with multiple non–real-time threads.

### 2.3.2. Interrupt Response

Embedded environments often require interaction with different external devices and interrupt response times for controllers because of the interaction in fields such as robots, industrial devices, and communication. Therefore, it is important to measure the interrupt latency when the Python-based real-time system is implemented.

Interrupt latency measurement experiments use GPIO on embedded boards, one as an external interrupt input pin and the second as an output pin for measuring the interrupt response time. A 1 kHz square wave reference signal is input to an interrupt pin set to detect the input pin's rising and falling edge. In a service routine executed by an interrupt caused by edge detection, enter the status value of the input pin as the control value of the output pin to measure the skew of the reference signal and the output pin as obtained

by the oscilloscope. Here, the skew of the two signal measurements refers to the interrupt response time.

Additionally, the experimental process is measured for five minutes, and the GPIO interrupt function is implemented using the wiringPi library. The results are compared and analyzed by implementing both C/C++ and Python.

Figure 4 shows the C/C++ and Python implementation interrupt responses. According to the skew results, Python averaged 24.225 μs, reaching up to 83.196 μs, and C/C++ averaged 20.640 μs, reaching up to 68.012 μs. The standard deviation is 3.756 μs for C/C++ and 3.782 us for Python, showing very similar deviations. Comparing each difference, 3.585 μs is shown at the average value, and 15.184 μs is shown at the maximum value, but when comparing Sdev, it can be seen as an outlier. As a result, the interrupt response shows similar performance between Python and C/C++.

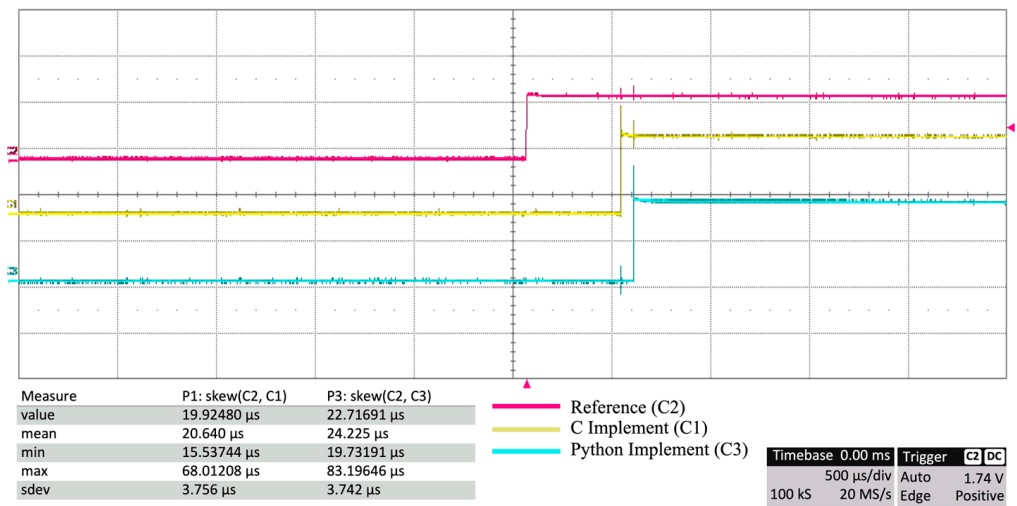

**Figure 4.** Interrupt response of C/C++ and python in the RT−Preempt patched Linux kernel.

### 2.3.3. Discussion

The feasibility of the Python real-time system implemented through this experiment was researched through periodicity and responsiveness, as well as preemption and interrupt responsiveness. It was confirmed that preemption and periodicity according to priority were observed, and there was no significant difference in interrupt response performance. However, in response performance, there were cases in which Python did not satisfy the WCRT predicted.

This result shows that Python is not satisfied with real-time systems with hard constraints, but it will be fully available in real-time systems with soft constraints. This paper can also serve as a springboard for flexible integration with existing studies of machine learning and numerical analysis on real-time systems for use in various areas, such as data analytics, robotics, and industrial control, which require execution and safety.

## 3. Experimental Validation

This chapter describes the experimental environment and method for verifying the feasibility of the implemented Python-based real-time control system and discusses the experimental results.

### 3.1. Environment

The environment used to implement the EtherCAT Master using a Python-based real-time system for this paper is shown in Table 5. The control unit uses the Intel MIO-5272U-U6A1E x86-based single board computer (SBC) used in industrial embedded systems and the low-power Intel i7-6600U processor, 8GB of DDR4 RAM, and an Intel i219 network controller with an e1000e driver.

**Table 5.** Python EtherCAT Master Development Environment.

| Item | Description |
| --- | --- |
| Master | |
| Board | MIO-5272U-U6A1E |
| CPU | Intel i7-6600U |
| Memory | DDR4 8GB |
| Network Controller | intel i219 |
| Linux Kernel | kernel 4.14.134-rt63 |
| OS Distribution | Lubuntu 18.04 |
| Python | 3.6.9 |
| EtherCAT Master | IgH EtherCAT Master 1.5.2 |
| Slave-1 | |
| Product | LS Mecapion L7N Servo Drive |
| PDO | 26 bytes for each slave (RxPDO 13 bytes, TxPDO 13 bytes) |
| Slave-2 | |
| Product | Beckhoff EL2024 Digital Output |
| PDO | 1 byte for each slave (RxPDO 1 byte, TxPDO 0 bytes) |

To build a real-time system environment for the EtherCAT Master using a Python-based real-time system on this platform, kernel 4.14.134-rt63, a stable version compatible with the EtherCAT master of the RT-PREEMPT-patched Linux kernel, is used. To ensure maximum real-time performance, the kernel uses a method suggested in previous studies to set kernel options for the power management and debugger sections [20]. Power management is an essential factor related to the latency of real-time systems and should be disabled.

Kernel debugging features are another source of latency that affects real-time requirements. In particular, KGDB functions, which are debuggers used to examine variables, the call stack information, and memory usage all affect the latency. Therefore, KGDB must be disabled. Similarly, Lubuntu is used to prevent real-time performance degradation due to the GUI and uses the latest Python version, 3.6.9, which can be applied accordingly.

The EtherCAT master uses the open-source IgH EtherCAT master, using version 1.5.2, which is known to be stable [21]. It also uses the Ctypes module to configure the IgH EtherCAT master shared library to interface with Python.

The EtherCAT slaves to be used to verify and test the implemented Python-based real-time system use LS Mecapion L7N servo drives and the Beckhoff EL2024 digital output. The PDO (process data object) used for communication is 13 bytes for transmission and reception for the servo devices and 1 byte for the digital IO.

This development environment can be an example of a real-time embedded system's platform selection and system environment configuration for controlling industrial machinery.

*3.2. Experimental Method*

To verify the operation and performance of the Python-based real-time control system implemented in this paper and apply it with EtherCAT, the periodicity, responsiveness, and synchronous tasks were measured using EtherCAT.

The application was executed through a real-time task of 1 ms using the Python-based real-time system and EtherCAT implemented in this paper. Python real-time tasks period and response time were measured. The period refers to the difference between the last start time and the current start time of the task, and the response time refers to the time from the task's current start time to EtherCAT transmission and reception and data processing. In addition, to reduce the overhead as much as possible when measuring the period and response time, these data were stored in a pre-generated NumPy array.

The configuration layout and actual images of the EtherCAT testbed used in the experiment are shown in Figures 5 and 6, respectively.

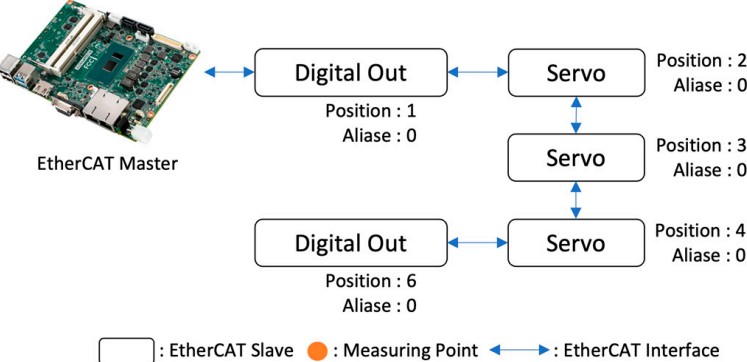

**Figure 5.** Environment diagram of the EtherCAT Master using Python-based real-time control system.

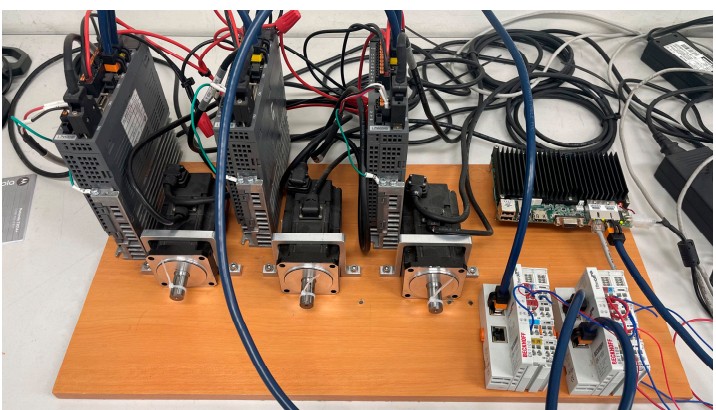

**Figure 6.** Environment picture of the EtherCAT Master using Python-based real-time control system.

The experiment used two digital outputs and three servos, with 1 byte for the digital outputs and 26 bytes for the servos. Data amounting to 80 bytes were exchanged during one control period.

The digital outputs (see Figure 5) executed the toggle control task every period and were configured at both ends of the EtherCAT network topology to measure the output pin of position 1 and the output pin of position 6 through an oscilloscope. For the servo, 1 Hz of sine control was performed up to −90 to 90 degrees, with the control being operated correctly. The experiment lasted a total of 5 min. It measured the period, response time, and position of the EtherCAT servo every 1 ms, which was the task period, and a total of 300,000 sample data were obtained. At the same time, the EtherCAT digital output pin was measured using an oscilloscope to measure the actual control period and synchronization of the EtherCAT slave device.

### 3.3. Experimental Results

The operation period and response time measurement results of the task described above are summarized in terms of statistical values, in this case, the mean, maximum (Max), minimum (Min), and standard deviation (Std), as shown in Table 6.

**Table 6.** Task Response time and Period Measurement Result.

| Metric | Response Time [ms] | Period [ms] |
| --- | --- | --- |
| Mean | 0.077 | 1 |
| Max | 0.131 | 1.238 |
| Min | 0.069 | 0.766 |
| Std | 0.004 | 0.001 |

As a result of the response time measurement, it was confirmed that the EtherCAT data transmission data processing and transmission operations were completed within the task operation period with a maximum of 0.131 ms, a minimum of 0.069 ms, and an average of 0.077 ms. In the periodic measurement results, the average operating period was 1 ms, but the maximum operating period was 1.238 ms, and the minimum period was 0.766 ms. This exceeded 1 ms, but this occurred once among 300,000 samples.

A histogram of the Z−scores calculated using Equation (2) to interpret the periodic measurement data probabilistic statistically is shown in Figure 7.

$$Z = \frac{\chi - \mu}{\sigma} \tag{2}$$

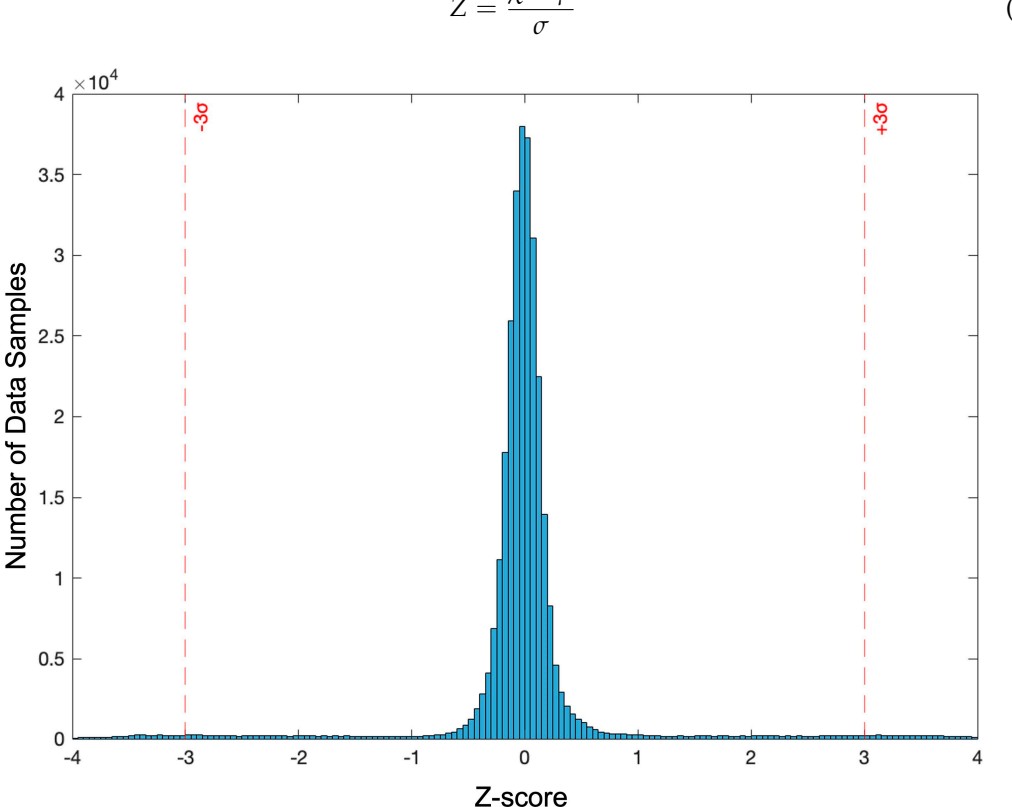

**Figure 7.** Histogram of the task period shown using Z−scores.

Through the calculations, the Z value can be used to determine the position of the measured data value in terms of the standard deviation. The ratio and probability of the normal distribution of the data can be determined by expressing these factors as a histogram [18].

The positions of the values 0, 1, 2, and 3 on the Z-score chart shown in Figure 7 have corresponding meanings identical to μ, σ, 2σ, and 3σ in a normal distribution, respectively. Thus, the Z-scores of −3 and 3 can be seen as −3σ and 3σ and −3σ and 3σ, respectively. It can also be seen that 99.7% of all data are distributed between these 3σ values, meaning that they are a good measure for determining the periodicity of real-time tasks. In addition, if the raw data of the Z-score −3 and 3 positions are obtained as shown in Equation (3), 99.7% of the total measurement period is distributed between 0.9961 ms and 1.0039 ms.

$$\chi = \mu + \sigma Z \tag{3}$$

Figure 8 shows the results after measuring the synchronicity of the two signals using an oscilloscope to locate the digital outputs at both ends of the actual hardware control period and the EtherCAT network topology.

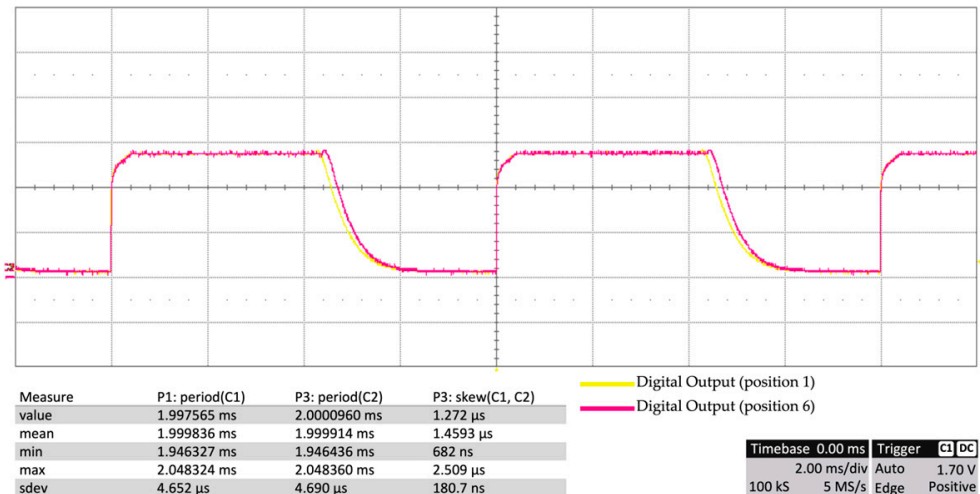

| Measure | P1: period(C1) | P3: period(C2) | P3: skew(C1, C2) |
|---------|---------------|----------------|------------------|
| value | 1.997565 ms | 2.0000960 ms | 1.272 µs |
| mean | 1.999836 ms | 1.999914 ms | 1.4593 µs |
| min | 1.946327 ms | 1.946436 ms | 682 ns |
| max | 2.048324 ms | 2.048360 ms | 2.509 µs |
| sdev | 4.652 µs | 4.690 µs | 180.7 ns |

Digital Output (position 1)
Digital Output (position 6)

Timebase 0.00 ms | Trigger C1 DC
2.00 ms/div Auto 1.70 V
100 kS 5 MS/s Edge Positive

**Figure 8.** Digital output control period and skew measurement results.

The digital output can be toggled for every 1 ms period, allowing the oscilloscope to predict a measurement period of 2 ms when taking measurements. The measurement results showed that the digital output of position 1 was measured and found to have an average period of 1.99 ms and a maximum period of 2.04 ms. The digital output of position 6 was also measured, showing an average period of 1.99 ms and a maximum period of 2.04 ms. Both signals had a standard deviation of 4.6 us and an expected toggle control period of 2.0 ms. In addition, skew measurements of the two signals showed that they had an average of 1.45 us, a maximum of 2.5 us, a minimum of 0.68 us, and a standard deviation of 0.18 us.

Figure 9 shows the result after measuring the position feedback data of three servo drivers for every 1 ms period, the control reference position at 2−3 s, and the position data of each servo driver.

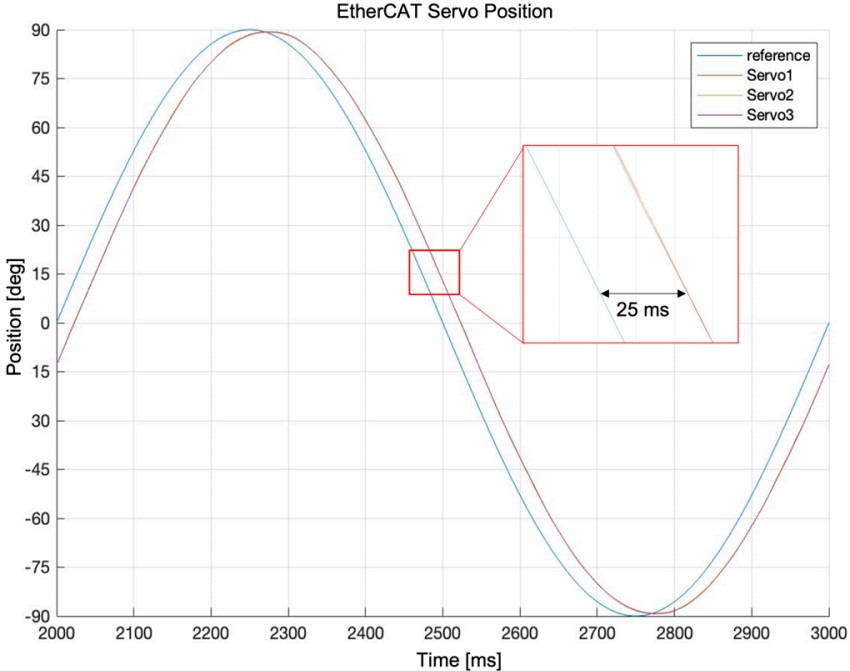

**Figure 9.** Servo position control feedback data measurement result.

The servo driver performed $-90$ to $90$ degree 1 Hz sine position control, as shown in Figure 9. When zoomed in around 2.5 s, the control response of approximately 25 ms is observable, indicating the performance of the position controller built into the servo driver. The results of the confirmed feedback data show that position control of the servo motor via EtherCAT was performed correctly.

As a result of the experiment, it was confirmed that precise control was possible through servo driver position control feedback data, with the performance of the implemented Python EtherCAT master verified by confirming an average simultaneity of 1.45 ms and the 1 ms constant period control. In addition, environments using motors and I/O, such as those in this experiment, can be used in applications such as conveyor belts, assembly processes, and industrial factories [22], with the use of Python-based programming languages enabling the flexible integration of technologies such as artificial intelligence and big data with EtherCAT control applications.

## 4. Conclusions

To demonstrate the feasibility of real-time systems in Python, we implemented real-time tasks in Python using the Ctypes module in a real-time Linux environment. We compared the real-time metrics of periodicity and responsiveness in Python and C/C++ languages. The statistical results always showed promising results in C/C++. While Python could be easily and rapidly implemented in the experimental condition, C/C++ had the disadvantage of being somewhat complicated and time-consuming concerning its coding and compiling processes. Lastly, an environment was built using EtherCAT Fieldbus-based servo motors and digital outputs used in actual industrial sites, and periodicity and synchronicity were measured through a python-based real-time control system experiment also performed statistical analysis using Z$-$score.

As a result, Python was shown to work in a real-time environment, task periodicity was satisfied, priority-based preemption occurred, and the feasibility of Python was researched on a real-time control system.

The results here can be applied to industrial controllers, robots, and the mobility field where real-time performance is required, and this method can extend the limits and relax the constraints of real-time performance with Python, enabling integration with real-time systems in various fields, such as artificial intelligence(AI), data science, and networking.

For future studies, as shown in RT$-$AIDE, performance evaluations and research using a greater variety of real-time metrics will be conducted to improve experiment and analysis methods, including power consumption, which may be applied only to RT-Preempt but also to several RTOSs, and both non$-$real-time and real-time types. In addition, we will be conducting research on intelligent control systems leveraging machine learning and neural network inference for service and industrial robots [23–25].

**Author Contributions:** S.Y.C. and R.D. contributed equally to this paper. The joint first authors surveyed the background of this research, conceptualized and developed the software and environment, formulated the experiment procedures, and analyzed the results of the experiments. B.W.C. supervised and supported this study. All authors have read and agreed to the published version of the manuscript.

**Funding:** This work has been financially supported by SeoulTech (Seoul National University of Science and Technology).

**Data Availability Statement:** The data presented in this study are available on request from the corresponding author.

**Conflicts of Interest:** The authors declare no conflict of interest.

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
