# Peer review of "Feasibility Study for a Python-Based Embedded Real-Time Control System"

_electronics, doi:10.3390/electronics12061426_

Round 1

Reviewer 1 Report

The paper proposes the development of a Python method for executing real-time periodical tasks under the real-time Linux extension RT-Preempt, using a POSIX Linker and the Ctypes module. Evaluation of the method on a Raspberry Pi 4 device through various real-time performance analyses and comparison of the results to the same program developed in C/C++ to validate the potential of the Python-based real-time system were conducted using Fieldbus EtherCAT.

*Few typo needs to be corrected such as ..experiments were conducted …” in abstract.

*It is a good idea and nice done. However, I would be more valuable describing the technical difficulty, especially what RT-Preempt cannot do.

*For systems used to Python programming, it might not be natural to convert to as periodic ones for real-time.

Author Response

  1. The paper proposes the development of a Python method for executing real-time periodical tasks under the real-time Linux extension RT-Preempt, using a POSIX Linker and the Ctypes module. Evaluation of the method on a Raspberry Pi 4 device through various real-time performance analyses and comparison of the results to the same program developed in C/C++ to validate the potential of the Python-based real-time system were conducted using Fieldbus EtherCAT.

Response: Thank you very much for your guiding comments. Each concern of the reviewer is detailed below which helped us improve the quality of the work. We are very hopeful that the revised manuscript has addressed all of these concerns for the successful publishing in the Electronics journal.

  1. *Few typo needs to be corrected such as “..experiments were conducted …” in abstract.

Response: We found wrong or awkward expressions in the Abstract and corrected them for the following reasons.

  • In the second sentence, "specified tasks should meet specific temporal deadlines" has been rephrased to "specific tasks must meet specific temporal deadlines" to improve clarity.
  • In the third sentence, realtime has been changed to “real-time” to maintain consistency throughout the paper. The phrase "period setting" is changed to "periodic" to make it more natural. We also clarified that the tasks are scheduled with fixed priorities and added the phrase “with fixed priorities”.
  • The fourth sentence changed to active voice from "experiments were conducted" to "we conducted experiments".
  • In the last sentence, "the feasibility of the proposed method" has been changed to "the feasibility of a Python-based real-time control system" to clarify what was being validated.
  1. *It is a good idea and nice done. However, I would be more valuable describing the technical difficulty, especially what RT-Preempt cannot do.

Response: Thank you for your feedback. We understand your suggestion to provide more detail on the technical difficulties and limitations of using RT-Preempt for real-time performance. However, our approach is not to overcome the limitations of RT-Preempt but facilitate Python to build real-time tasks on top of RT-Preempt as shown in Fig. 1. Finally, we will consider another approach to building Python-based real-time tasks on top of other native RTOSs for future works.

  1. *For systems used to Python programming, it might not be natural to convert to as periodic ones for real-time.

Response: Python's global interpreter lock and garbage collector can introduce delays and make it sometimes challenging to ensure real-time performance. However, it needs to be more accurate to suggest that systems used in Python programming are inherently unsuited for real-time applications.

Several approaches and tools are available for developers to achieve performance in Python programming. However, we are developing real-time systems where the periodic task is useful for utilizing a fixed-priority preemptible scheduler and implementing real-time tasks in Python.

In conclusion, we think that while Python might have some limitations regarding real-time applications, it is possible to achieve real-time performance in Python with the right approaches, as suggested in our paper.

Reviewer 2 Report

According to the authors the paper describes a Python method for executing real-time periodical tasks under the real-time Linux extension RT-Preempt using a POSIX Linker and the Ctypes module. The method is tested on a Raspberry Pi 4 device and evaluated through various real-time performance analyses, including periodicity and interrupt latency. The results are compared to the same program developed in C/C++. The paper also demonstrates the feasibility of a Python-based real-time embedded control system by performing servo motor control using the industrial Fieldbus EtherCAT.

Among others, the authors highlight one of the significant contributions of the work a comparison of the results with the same program developed in C/C++ for validation the potential of a real-time system based on Python.

The authors use the CTypes library that allows Python to communicate with C/C++ from a MS Windows DLL or UNIX/Linux SO (Shared Object ) files. DLL is an executable file, which performs certain functions that cannot be run by itself but is run from an executable executable files.

In the paper, the authors use Python as a "trigger" that will trigger activities that are already written in C/C++. So, in fact, the authors measure and document the execution speed of C/C++ code run by Python, that is, Python is just a "caller" requesting the execution of C/C++ code. In the paper, the authors compare the execution speed of C/C++ code and the execution speed of C/C++ code called by Python.

All in all, it is a review paper that is well constructed, methodologically sound and set, described and supported by experiment as evidence. However, there is a lack of scientific contribution or exceptional novelty that highlights this work as important and stimulating for the entire Python scientific community.

In this form paper did not particularly contribute to revealing and bringing Python as a popular programming languages ​​and directing and recommending for use in applications of systems controlling.

Simply, more is expected in the title and in the Abstract from authors who did not confirm what was stated or promised in the continuation of the work.

The work has a few small typfelers and mislabeled Figure 5 instead of Figure 7 in text line number 370.

The list of literature on the use of Python and the use of similar applications for system controlling is too small and insufficient for this topic.

Author Response

  1. According to the authors the paper describes a Python method for executing real-time periodical tasks under the real-time Linux extension RT-Preempt using a POSIX Linker and the Ctypesmodule. The method is tested on a Raspberry Pi 4 device and evaluated through various real-time performance analyses, including periodicity and interrupt latency. The results are compared to the same program developed in C/C++. The paper also demonstrates the feasibility of a Python-based real-time embedded control system by performing servo motor control using the industrial Fieldbus EtherCAT.

Among others, the authors highlight one of the significant contributions of the work a comparison of the results with the same program developed in C/C++ for validation the potential of a real-time system based on Python.

Response: We are very appreciative that the reviewer has clearly understood the logical flow of the paper and the main goal of presenting the feasibility of Python-based real-time system as a feasible alternative to low level language such as C/C++.

  1. The authors use the CTypes library that allows Python to communicate with C/C++ from a MS Windows DLL or UNIX/Linux SO (Shared Object ) files. DLL is an executable file, which performs certain functions that cannot be run by itself but is run from an executable executable files.

In the paper, the authors use Python as a "trigger" that will trigger activities that are already written in C/C++. So, in fact, the authors measure and document the execution speed of C/C++ code run by Python, that is, Python is just a "caller" requesting the execution of C/C++ code. In the paper, the authors compare the execution speed of C/C++ code and the execution speed of C/C++ code called by Python.

Response: We affirm that we have used the CTypes library to wrap a C/C++ API to run real-time threads in Python. However, we clarify that if no native Python code is within the callback function, the reviewer is correct that it would only by “caller” that triggers C/C++ code. But in our implementation and during the experiments, we have used a simple print function to ensure interpreter response.

  1. All in all, it is a review paper that is well constructed, methodologically sound and set, described and supported by experiment as evidence. However, there is a lack of scientific contribution or exceptional novelty that highlights this work as important and stimulating for the entire Python scientific community. In this form paper did not particularly contribute to revealing and bringing Python as a popular programming languages ​​and directing and recommending for use in applications of systems controlling. Simply, more is expected in the title and in the Abstract from authors who did not confirm what was stated or promised in the continuation of the work.

Response: We are very appreciative of the opinion of the reviewer and accept constructive criticisms in this round of review. To clarify the contributions of this work, we have organized the contributions and the necessary details to clarify them in the manuscript. These are:

  • The development of a Python method for executing real-time periodical tasks under the real-time Linux extension RT-Preempt, using a POSIX Linker and the Ctypes module.
  • Evaluation of the method on a Raspberry Pi 4 device through various real-time performance analyses, including periodicity and interrupt latency.
  • Comparison of the results to the same program developed in C/C++ to validate the potential of the Python-based real-time system.
  • Demonstration of the feasibility of a Python-based real-time embedded control system by performing servo motor control using the industrial Fieldbus EtherCAT.
  • Providing a stepping-stone for easier integration of recent trends in machine learning and numerical analysis on real-time systems for use in various fields such as data analytics, robotics, and industrial control.

We are more focused on evaluating the performance of Python and presenting its feasibility compared to the conventional method of using C/C++ to implement real-time applications. We clarify that our approach does not claim that the Python approach is better than any other programming language. But that it could be another and easier solution to build real-time systems. The experimental results have shown that the performance of Python is very good for real-time systems.

Also, to evaluate the proposed method in a realistic environment, we have added Table IV on Page 4, which shows the experimental results in an environment where multiple non real-time threads run alongside real-time tasks. To emulate this environment, we have performed the same experiment method but with the injection of best-effort interfering loads (CPU and memory stress). Multiple non real-time threads attempt to occupy the CPU and memory resources when available. This test should have minimal effect on the performance of real-time tasks with high priorities. Otherwise, it can be concluded that real-time tasks violate real-time constraints, which may lead to missing stringent deadlines. Our results have shown that the average period of all real-time tasks is equal to the idle environment with a minimal increase in the standard deviation. The response time has also increased slightly compared to the results in an idle environment. Despite their increased response times, all real-time tasks have met their deadlines. This means that Python-based real-time tasks with high priorities are scheduled as expected and can satisfy real-time constraints even in an environment with multiple non real-time threads.

  1. The work has a few small typfelers and mislabeled Figure 5 instead of Figure 7 in text line number 370. The list of literature on the use of Python and the use of similar applications for system controlling is too small and insufficient for this topic.

Response: The typing error on line 370 you pointed out was found and corrected. Also, this paper was updated in English by a native speaker of Seoul National University of Science and Technology in Korea, and we also checked and revised it. Still, we reviewed it again to proceed with the improvement. Regarding the list of literature, we have cited current efforts and similar researches that addresses the real-time problem using Python.

We hope the revised manuscript has addressed the reviewer's concern and may find its improvement worthy of publishing in the Electronics journal.

Reviewer 3 Report

1.      The main question addressed by the research is: is it possible to construct an embedded real-time control system using Python as a programming language.

2.      The topic is relevant in the field because it addresses the demand for more flexible and efficient programming languages for real-time control systems. It also fills a vacuum in the area by investigating the viability of utilizing Python for embedded real-time control systems.

3.      The study adds to the subject area by offering a feasibility analysis for employing Python in embedded real-time control systems, which has not been extensively researched in the literature. The paper also compares Python and C, which might be valuable for both scholars and practitioners in the field.

4.      The methodology of the study is generally sound, but there are some areas for improvement. For example, the authors should consider including more information about the hardware and software platforms utilized in the study. They can also think about incorporating more statistical analysis to support their conclusions. Other controls, such as comparing Python to other programming languages used in real-time control systems, should be considered..

5.      The conclusions are consistent with the evidence and arguments presented and do address the main question posed. The authors provide a thorough analysis of the feasibility of using Python for real-time control systems and discuss the advantages and limitations of using Python compared to C.

6.      The references used in the study are generally appropriate and relevant to the topic. The authors cite a range of sources from peer-reviewed journals and other academic publications.

7.      The tables and figures in the document are clear and well-presented, and provide useful visual aids to help readers understand the data and results presented in the study.

The research takes a preliminary look at the viability of utilizing Python for embedded real-time control systems. While the findings indicate that Python may be utilized for such systems, there are constraints that must be addressed, and further study is required to properly assess Python's potential in this context. As a result, the study provides a platform for future research.

Author Response

  1. The main question addressed by the research is: is it possible to construct an embedded real-time control system using Python as a programming language.

Response: We very much appreciate that the reviewer has completely understood the main problem addressed by this work.

  1. The topic is relevant in the field because it addresses the demand for more flexible and efficient programming languages for real-time control systems. It also fills a vacuum in the area by investigating the viability of utilizing Python for embedded real-time control systems.

Response: We agree with the reviewer that there is a research gap in the field of using higher level language to implement real-time control systems. We are very optimistic that this work will be a good stepping stone for further research in this field.

  1. The study adds to the subject area by offering a feasibility analysis for employing Python in embedded real-time control systems, which has not been extensively researched in the literature. The paper also compares Python and C, which might be valuable for both scholars and practitioners in the field.

Response: Thank you for this very optimistic comment that appreciate the future application of this work both in the academe and industry.

  1. The methodology of the study is generally sound, but there are some areas for improvement. For example, the authors should consider including more information about the hardware and software platforms utilized in the study. They can also think about incorporating more statistical analysis to support their conclusions. Other controls, such as comparing Python to other programming languages used in real-time control systems, should be considered.

Response: We appreciate your suggestions to include more information about the hardware and software platforms used in our study, as well as to incorporate more statistical analysis and compare Python to other programming languages used in real-time control systems. We will consider these points for future research.

  1. The conclusions are consistent with the evidence and arguments presented and do address the main question posed. The authors provide a thorough analysis of the feasibility of using Python for real-time control systems and discuss the advantages and limitations of using Python compared to C.
  2. The references used in the study are generally appropriate and relevant to the topic. The authors cite a range of sources from peer-reviewed journals and other academic publications.
  3. The tables and figures in the document are clear and well-presented, and provide useful visual aids to help readers understand the data and results presented in the study.

The research takes a preliminary look at the viability of utilizing Python for embedded real-time control systems. While the findings indicate that Python may be utilized for such systems, there are constraints that must be addressed, and further study is required to properly assess Python's potential in this context. As a result, the study provides a platform for future research.

Response: For the rest of the comments which are very related to each other. Our response is as follows: We are much honored that the reviewer appreciated and completely understood the contributions of this paper. Thank you for your insightful feedback.

We are glad that you found our conclusions consistent with the evidence presented and that we provided a thorough analysis of the feasibility of using Python for real-time control systems. We will continue improving the paper, considering your suggestions and other feedback we receive. Thank you again for your valuable input.

Reviewer 4 Report

This paper presents a Python method for executing periodic real-time tasks under the fixed-priority preemptible scheduler of RT-Preempt and the feasibility of a Python-based real-time embedded control system by performing servo motor control using the industrial Fieldbus EtherCAT, as a high-speed Ethernet to a dynamic servo drive.

The paper is interesting, well structured and well written.

The clarity of Figures 2, 3, 4, and 8 should be improved.

Author Response

We are much honored that the reviewer appreciated and completely understood the contributions of this paper. We are very grateful for your nice comments. Figures 2, 3, 4, and 8 have all been modified, and the revised figures are highlighted in red in the manuscript. Since it is a captured image from an oscilloscope, the clarity may be somewhat reduced, but we have improved it, as shown in the images below, to make it as clear and sharp as possible.

Round 2

Reviewer 2 Report

According to the revised and improved text submitted, it is still a review paper with an insufficient list of literature and a low level of scientific importance. The purpose of Python is to use a wide base of functions and very rich files that mainly serve the programmers to facilitate the execution of complex mathematical operations in a simpler and more convenient way. Of course, execution speed is realistically the most problematic parameter.

Open access databases are full of examples of using Python for various management purposes. I advise you to refer to some of them to improve the reference, comparability and traceability of this review paper.

Author Response

According to the revised and improved text submitted, it is still a review paper with an insufficient list of literature and a low level of scientific importance. The purpose of Python is to use a wide base of functions and very rich files that mainly serve the programmers to facilitate the execution of complex mathematical operations in a simpler and more convenient way. Of course, execution speed is realistically the most problematic parameter.
Response: We agree with the reviewer that complex mathematical operations may result to issues with execution speed. But we would also like to clarify that although the execution time may be relevant in applications with heavy computational load, our research is more focused on addressing the real-time constraints with Python, which is to ensure that tasks can respond to events under stringent time constraints (deadline). 

Also, to the best of our knowledge, review papers are more focused on collecting and studying a list of literature with the aim to present readers the insight of the authors about a certain topic. However, our work (as in the title), we have presented a feasibility study of Python-based real-time systems. As we have listed in the contributions in bullet form in the introduction. We would also like to note that we have not encountered any issue with the form of submission (regular article) among all of the reviewers.

•    We have developed a Python-based method to execute real-time periodic tasks under the real-time Linux extension RT-Preempt, using a POSIX Linker and the Ctypes module.
•    We have  evaluated the proposed method on a Raspberry Pi 4 device through various real-time performance analyses, including periodicity and interrupt latency.
•    Comparison of the results to the same program developed in C/C++ to validate the potential of the Python-based real-time system.
•    Demonstration of the feasibility of a Python-based real-time embedded control system by performing servo motor control using the industrial Fieldbus EtherCAT.

We would defer that our paper has an insufficient list, rather that there is a small effort in the community (academe or industry) regarding Python-based real-time systems. Although we agree that our work may lack high-level theoretical contribution algorithmic-wise, our work and the results are valuable in the practical-sense where devices should be controlled in a safety-critical manner. Also, one of our aims to provide future readers and researchers a stepping-stone for easier integration of recent trends in machine learning and numerical analysis on real-time systems for use in various fields such as data analytics, robotics, and industrial control. 

Open access databases are full of examples of using Python for various management purposes. I advise you to refer to some of them to improve the reference, comparability and traceability of this review paper.
Response: As in the previous response, we would like to clarify that with our main contributions, we have not encountered any issues with the form of this submission, which is a regular article. But we adhere to the reviewer to improve the reference regarding Python, especially regarding the available datasets for application in machine learning. Thus, we have added the following as reference [23-25] in the revised manuscript.

  1. Yasar, M.S.; Evans, D.; Alemzadeh, H. Context-Aware Monitoring in Robotic Surgery. In Proceedings of the 2019 International Symposium on Medical Robotics (ISMR); IEEE, April 2019; pp. 1–7.
  2. Li, Z.; Hutchinson, K.; Alemzadeh, H. Runtime Detection of Executional Errors in Robot-Assisted Surgery. In Proceedings of the 2022 International Conference on Robotics and Automation (ICRA); IEEE, May 23 2022; pp. 3850–3856.
  3. Long, Y.; Wu, J.Y.; Lu, B.; Jin, Y.; Unberath, M.; Liu, Y.-H.; Heng, P.A.; Dou, Q. Relational Graph Learning on Visual and Kinematics Embeddings for Accurate Gesture Recognition in Robotic Surgery. In Proceedings of the 2021 IEEE International Conference on Robotics and Automation (ICRA); IEEE, May 30 2021; pp. 13346–13353.